# Sodium Benzoate—Harmfulness and Potential Use in Therapies for Disorders Related to the Nervous System: A Review

**DOI:** 10.3390/nu14071497

**Published:** 2022-04-02

**Authors:** Łucja Justyna Walczak-Nowicka, Mariola Herbet

**Affiliations:** Chair and Department of Toxicology, Faculty of Pharmacy, Medical University of Lublin, 20-090 Lublin, Poland; lucja.wn18@gmail.com

**Keywords:** conservatives, food additives, sodium benzoate, neurodegenerative diseases, depression

## Abstract

Currently, due to the large number of reports regarding the harmfulness of food additives, more and more consumers follow the so-called “clean label” trend, i.e., prefer and choose the least-processed food products. One of the compounds known as a preservative with a high safety profile is sodium benzoate. While some studies show that it can be used to treat conditions such as depression, pain, schizophrenia, autism spectrum disorders, and neurodegenerative diseases, others report its harmfulness. For example, it was found to cause mutagenic effects, generate oxidative stress, disrupt hormones, and reduce fertility. Due to such disparate results, the purpose of this study is to comprehensively discuss the safety profile of sodium benzoate and its potential use in neurodegenerative diseases, especially in autism spectrum disorder (ASD), schizophrenia, major depressive disorder (MDD), and pain relief.

## 1. Introduction

Due to the advancing chemicalization of food in recent years, an increasing number of consumers have declared their interest in such food features as sensation, health, and, above all, safety [1,2,3,4,5]. For fear of the adverse effects of chemicals added to food in order to improve its taste and appearance or extend its shelf life, the choice of the least-processed products that do not contain additives, including preservatives, has become more and more popular, thus creating the trend of the so-called “clean label”.

Sodium benzoate (according to the European nomenclature E211) is a salt of benzoic acid and is well soluble in water, tasteless, and odorless, and due to its antifungal and antibacterial properties, it is a preservative added to food in strictly defined doses. It inhibits the growth of bacteria, yeast, and mold [6]. Sodium benzoate was approved as the first of all food preservatives by the Food and Drug Administration (FDA). The permissible limit of its consumption is 0–5 mg/kg of body weight. It also has a GRAS (generally regarded as safe) status according to the FDA [7]. Sodium benzoate is considered safe for human health if it is consumed in amounts of less than 5 mg/kg of body weight per day. At this level, the Acceptable Daily Intake (ADI) was established. It determines the dose of a given substance that can be consumed by a person daily throughout his or her life without suffering any health damage.

Sodium benzoate does not accumulate in the body. Benzoate is conjugated with glycine to form hippurate in the liver and kidney in a reaction occurring in the mitochondrial matrix [8]. Upon entering the matrix, the compound is converted to benzoyl-coenzyme A (CoA) (ligase) and then to hippurate (glycine N-acyltransferase), which leaves the mitochondrion. It is excreted primarily through the urinary system. The administration of sodium benzoate causes a strong but transient increase in anthranilic acid (involved in tryptophan metabolism) and acetylglycine. The benzoate is one of the cinnamon metabolites [9,10]. Cinnamon contains cinnamaldehyde, which is converts to cinnamic acid in the liver and is then β-oxidized to benzoate (sodium salt or benzoyl-CoA). It is easily absorbed from the gastrointestinal tract and metabolized in the liver into hypuronic acid. In this form, it is excreted from the body with urine usually within 6 h of ingestion. 

Due to its properties, sodium benzoate is used to preserve food products with an acidic pH, such as fruit pulp and purees, jams, pickles, pickled herring and mackerel, margarine, olives, beer, fruit yogurts, canned vegetables, and salads [11]. Most often, sodium benzoate is added to carbonated drinks, sauces, mayonnaises, margarines, tomato paste, and fruit preserves. In turn, in its natural form, it is present in, among other things, cinnamon, mushrooms, cranberries, blueberries, and cloves. Therefore, sodium benzoate is classified as a compound with a broad safety profile. It is also approved for therapeutic use in the form of two drugs: Ammonul/Ucephan and Buphenyl [12,13]. The indications for their use are urea cycle disorders, as well as hyperammonemia. Moreover, some studies report that not only is sodium benzoate an excellent preservative, but it may also have potential therapeutic uses in the treatment of diseases such as major depressive disorder (MDD), schizophrenia, autism spectrum disorder (ASD), and neurodegenerative diseases. Officially, sodium benzoate is regarded as not harmful—only when consumed in large amounts can it cause allergic reactions or contribute to the exacerbation of disease symptoms in aspirin-induced asthma (with hypersensitivity to aspirin and other non-steroidal anti-inflammatory drugs) [14,15,16]. However, in recent years, some reports have provided for its adverse health effects. Interestingly, various and often contradictory results of scientific research prove that sodium benzoate has unfavorable or, on the contrary, beneficial effects on the body (especially in the treatment of certain diseases) by engaging in the same mechanisms of action. Due to the controversial and often contradictory results of reports and studies, the aim of this study is to assess on the one hand the adverse effects of sodium benzoate and on the other its potential use in the selected diseases related to the nervous system.

## 2. The Harmfulness of Sodium Benzoate

It is believed that benzoate can be transformed by decarboxylation into toxic benzene, especially in combination with vitamin C, and then become a compound of high toxicity, mutagenicity, and teratogenicity [17]. There are also reports that sodium benzoate has a weak genotoxic effect. Moreover, it was shown to increase the DNA damage in human lymphocytes in vitro. The compound did not affect the rate of replication, but it did reduce the mitotic rate [18]. Mutagenic and genotoxic effects were also demonstrated in another study on human lymphocytes [19]. This compound caused micronucleus formation and chromosome breakage. In addition, the research shows that sodium benzoate generates oxidative stress and has an adverse effect on the immune system, liver, kidneys, and fertility.

### 2.1. The Effect of Sodium Benzoate on the Oxidative Stress and Inflammation

Oxidative stress, but also the inflammatory process, play an important role in the pathomechanisms of many diseases. Oxidative stress is associated with an imbalance between the production of reactive oxygen species (ROS) and the amount of antioxidants or radical scavengers [20]. The excess of oxygen free radicals leads to the activation of transcription factors such as NF-κB, AP-1, and HIF-1α, as well as pro-inflammatory genes, leading in turn to the induction of proteins, cell adhesion molecules (CAM), monocyte chemoattractant protein (MCP-1), tumor necrosis (TNF-α), interleukin (IL) -1, and transforming growth factor (TGF-β). ROS may also affect tyrosine kinases, including Src, Ras, phosphoinoside 3-kinase (PI3K), epidermal growth factor receptor (EGFR), mitogen-activated protein kinase (MAPK) or p38MAPK, c-Jun-N-terminal kinase (JNK), and extracellular signal-regulated kinases (ERK) by inducing inflammatory processes and aging mechanisms. On the other hand, the ongoing inflammatory process leads to an increase in oxidative stress, which in turn creates a vicious circle [21]. Moreover, according to the free radical theory of aging, the accumulation of oxidative damage leads to the loss of cell functionality, which in turn leads to cell death [22]. It has therefore been suggested that oxidative stress and related inflammatory processes are related to age-related diseases. Increased levels of oxidative stress and inflammatory processes are observed, inter alia, in neurodegenerative diseases, cancer, diseases of the bile ducts and kidneys, diabetes, and cardiovascular diseases [23]. The effect of sodium benzoate (6.25, 12.5, 25, 50, and 100 μg/mL) on the increasing oxidative stress was observed in erythrocytes in an in vitro study [24]. After the treatment of cells with benzoate, there was observed increased lipid peroxidation, as well as decreased levels of antioxidant enzymes, such as superoxide dismutase (SOD), catalase, and glutathione S-transferase. In another study, its effect on the induction of apoptosis was observed [25]. In addition, inhibition of antioxidant enzymes, decreased levels of glutathione (GSH), increased levels of nitric oxide (NO), and inflammation (increased in IL-6 and TNF α) were noted. The effects of sodium benzoate on oxidative stress have been also examined on male rats at different doses and for different periods of time [26]. In some of the study groups, decreased levels of GSH and malondialdehyde (MDA) were observed. Sodium benzoate also affected sodium and potassium levels (elevation). The effect of benzoate on oxidative stress was associated with an oxidative damage increase. This effect was confirmed in another study [27]. Sodium benzoate was administered to rats for 30 days at different doses (70, 200, 400, and 700 mg/kg b.w.) [28]. In this study, the dose of 70 mg/kg was found to be safe; however, at higher doses, the compound decreased the antioxidant enzymes activity.

The effect of benzoate on B and T lymphocyte reactivity was also investigated [29]. The highest non-cytotoxic dose of the compound (1000 mg/mL), determined by the MTT assay was chosen for the study. This dose decreased the functional activity of both classes of lymphophytes. Additionally, sodium benzoate affected the cell cycle by stopping the cells in the G1 phase. It also inhibited T lymphocyte proliferation against allogeneic MHC antigens and affected cytokine levels. There was a decrease in CD8 expression for T lymphocytes and CD19 for B lymphocytes and in the expression of activation markers such as CD95 (both classes of lymphocytes), CD28 (T lymphocytes), and CD40 (B lymphocytes). In vivo studies showed that administrating sodium benzoate (200, 400, and 700 mg /kg b.w) to rats for 30 days increased the level of pro-inflammatory cytokines (TNF-α, IFN-γ, IL-1β, and IL-6) and decreased the body weight of the animals [28].

### 2.2. Effect of Sodium Benzoate on the Embryos

A teratogenic study on sodium benzoate was reported in a zebrafish model [30]. At low doses (1–1000 ppm), the embryos exhibited a 100% survival rate, but higher doses caused deformation of the larvae. Another study based on the same model also showed that embryo survival depended on the time and dose [31]. The larvae were also characterized by reduced locomotor activity and decreased expression of tyrosine hydroxylase and dopamine transporter. In another study, it was reported that the effect of benzoate may be cumulative when it affects hormones [26]. It is possible that such an effect is also noted for other parameters. Moreover, some experiments were conducted on pregnant female rats administered with sodium benzoate (0.5, 1, and 1.5 mg/mL) [32]. This compound had little effect on maternal weight gain, but no toxic effects were observed. Perinatal mortality was significantly increased in the 1% and 1.5% benzoate dose groups. However, no fetal malformations or weight loss were observed. Benzoate also showed genotoxic effects on liver tissue in both fetuses and mothers. In contrast, in another study, sodium benzoate (doses 9.3 and 18.6 mmol/kg b.w.) decreased the fetal weight of rats and increased their mortality [33]. In addition, one more study found fetal deformities after prior treatment of pregnant females with benzoate (280 and 560 mg/kg b.w.) [34]. Among them, the following were observed: skin hemorrhages, craniofacial deformities, limb defects, spine defects, and neural tube defects. Developmental defects in mouse fetuses were also reported in a study where potassium benzoate was administered (280 and 560 mg/kg b.w.) [35]. Eye development defects, such as deformed lenses and also retinal folds with undeveloped layers accompanying hemorrhages, were observed in these fetuses. Another study also confirms its harmful effects on the fetus (280 and 560 mg/kk b.w.) [36]. Contrasting results regarding the teratogenicity of benzoate were observed in chickens (5–200 mg/kg b.w.) [37]. This compound did not adversely affect neural tube development in these embryos. However, its use should be limited, especially in pregnant women, due to its potential teratogenic properties.

### 2.3. Effect of Sodium Benzoate on Hormone Levels

It should be noted that benzoate was shown to affect the sperm motility (1 mg/kg b.w./day) [25]. It caused changes in the reproductive organs and affected the levels of sex hormones. Moreover, in another study, sodium benzoate affected the male reproductive system. The compound caused a 50% reduction in sperm count compared to the control group, as well as increased oxidative stress [38]. Another study also reported testicular dysfunction in rats after the administration of sodium benzoate at 100 mg/kg of body weight for 28 days [39]. This was associated with impaired semen quality and endocrine function of the testes and changes in their structure. In another in vivo study, the compound (0.01 mg/kb b.w.) also affected sex hormone levels (follicle-stimulating hormone (FSH), luteinizing hormone (LH), and free testosterone) [40]. A similar observation was related to the decrease in FSH, LH, and testosterone levels (sodium benzoate: 280 mg/kg/day) [41]. Additionally, a decrease in thyroxine and triiodothyronine and an increase in thyrotropin were observed. In contrast, another study reported a decrease in both thyroxine and thyrotropin (the dose of sodium benzoate was 50–200 mg/kg/day) [42]. 

### 2.4. Effect of Sodium Benzoate on Liver and Kidney Function

The sodium benzoate in animals affected the lipid profile and liver and kidney parameters. Moreover, there were observed histopathological and dose-dependent changes in the biochemical markers of liver damage (150–700 mg /kg b.w) [28,43]. It affected the histology of the kidney and liver. In addition, it was found that sodium benzoate may rather affect the kidneys than the liver [44]. This compound (100 mg/kg bw) was added to drinking water for 15 weeks. Similar to the previous study, the rats showed histological changes, including necrosis and atrophy of glomeruli and tubules, as well as increased urea and creatinine and decreased antioxidant defense. Another in vivo study confirmed its negative effects on the liver, as evidenced by an increase in the serum liver enzymes (alkaline phosphatase, aspartate aminotransferase (AST)) (the doses of sodium benzoate were 30, 60, 120 mg/kg b.w./day) [45]. In a rat model of schizophrenia, benozesan also affected liver parameters, i.e., alanine transaminase and AST increased and total protein and albumin decreased [40]. Furthermore, the toxic effects of benzoate (sodium benzoate administered 2.40% (for rats) and 3.00% (for mice) of sodium benzoate) on the liver were observed in F344 rats and B6C3F1 mice [46]. In the test animals, changes in the liver parameters, i.e., albumin, total protein, γ-glutamyltranspeptidase, and elevated serum phospholipids and cholesterol were observed. Sodium benzoate also caused an increase in the absolute liver weight in both animal species and in the absolute kidney weight of the rats. As in the previous studies, histological changes were observed. Negative effects on the liver and kidney were also confirmed in other experiments [47,48,49].

### 2.5. Sodium Benzoate and Children’s Hyperactivity

Attention-deficit hyperactivity disorder (ADHD) is mainly associated with symptoms of hyperactivity, inattention, and impulsivity [50]. Beverages containing benzoate preservatives in their composition were given (45 mg/day) to 3-year-old children, who then experienced an increase in hyperactivity [51]. These behaviors were reduced after the withdrawal. Another study showed a similar effect of sodium benzoate in 8-, 9-, and 3-year-old children [52]. Furthermore, a survey was conducted among college students that examined the association between the consumption of sodium benzoate-rich beverages and symptoms associated with ADHD [53]. Thus, it was shown that the consumption of such beverages was associated with a higher prevalence of symptoms of ADHD. However, it should be noted that due to the nature of this study (i.e., a survey), the feelings of the respondents were subjective.

The association of sodium benzoate with ADHD symptoms may be supported by the reports regarding its effects on the HPA axis [26]. In some subtypes of disease, there is hyperactivity of this axis [54,55]. In addition, it has been mentioned previously that benzoate consumption may be associated with increased allergy tendencies [56]. This is another feature that links sodium benzoate to ADHD. Allergy prevalence is shown to be associated with the occurrence of this [57,58]. In some studies, sodium benzoate was adversely affected by oxidative stress and inflammation, as described above. It is therefore highly probable that the consumption of benzoate-rich beverages is associated with the occurrence of ADHD. In such patients, antioxidant defense is impaired, and inflammatory parameters are increased [59,60]. They also develop a decrease in cholesterol in the LDL fraction [61]. As mentioned above, sodium benzoate lowers the cholesterol [62], and such a property might also exacerbate the ADHD symptoms. Furthermore, it was shown that the cerebellar dysfunction may play an important role in the pathogenesis of ADHD [63]. It is noteworthy that sodium benzoate decreased the volume of the hemispheres, cortex, and intracerebral nuclei and the number of cells in the cortex in rats [64]. In ADHD patients, sodium benzoate may have some harmful effects. Consumption of benzoate-rich products should also be avoided among these patients.

### 2.6. Sodium Benzoate—Irritating Effect on the Gastric Mucosa

The effect of benzoate (oral provocation with 20 mg of sodium benzoate) on gastric mucosa was studied in a clinical trial [65]. It was shown that it increased the release of allergic mediators, i.e., histamine and prostaglandins, from the mucosa compared to the control group. The same study suggested that benzoate-related allergic reactions may be mediated by prostacyclins and histamine.

### 2.7. Sodium Benzoate with Vitamin C

Although there are no studies that clearly confirm the harmfulness of these additives, it has been proven that when used as a preservative, sodium benzoate can react with vitamin C and thus form carcinogenic benzene [17]. In practice, this combination is often used in colorful, sweetened drinks. In many studies, elevated levels of benzene were reported in carbonated beverages, fruit juices, and other products where benzoate was present in combination with vitamin C [66,67,68,69]. It has been shown that the hydroxyl radical, formed by the metal-catalyzed reduction in O_2_ and H_2_O_2_ by ascorbic acid, can attack benzoic acid to form benzene [70]. However, it is worth noting that heat and light can increase the rate of benzene formation [71]. In one in vivo study, ascorobic acid exacerbated the deleterious effects of sodium benzoate on fertility [39]. Among other things, it potentiated damage to testicular tissue structure and deterioration of semen quality induced by benzoate.

### 2.8. Effects of Sodium Benzoate on Memory and Anxiety Processes

Wistar (healthy) rats received sodium benzoate in different concentrations in water (from 0.5–2%) [72]. This compound was shown to increase anxiety-like, depressive, and antisocial behaviors. Similar results were observed in another study where rats were given benzoate at a dose of 200 mg/kg/day [73]. Animals treated with these compounds experienced an increase in anxiety-like behaviors and impaired motor skills. The researchers suggest that this may be related to decreased levels of glycine in the body (it is consumed as a result of benzoate detoxification) and disruption of processes affected by this amino acid or disruption of zinc levels.

In another study performed on rats treated with sodium benzoate, an increase in body weight and food intake and a decrease in memory scores and anxiogenic effects were reported [74]. In addition, an increase in brain MDA, acetylcholinesterase (AChE), and caspase 3 and a decrease in TNF-α and IL-10 were observed. There was also an increase in SOD levels at 125 mg/kg and 250 mg/kg and a decrease in SOD at 500 mg/kg. Thus, it has been shown that benzoate enhances inflammatory response and has proapoptotic effects.

Sodium benzoate was also tested in a zebrafish (larva) model [75]. The compound-induced developmental defects in them and also contributed to increased oxidative stress and the development of anxiety-like behaviors. The developmental defects that were most commonly observed were yolk sac edema, pericardial edema, and tail-bending. In addition, it should be noted that larval mortality was observed even at lower than acceptable concentrations (<1000 ppm).

In another study, sodium benzoate (0.56, 1.125, and 2.25 mg/mL) was administered to healthy mice in different concentrations for 4 weeks [76]. It was shown that in healthy animals, benzoate induced impaired memory and motor coordination compared to the control group. Benzoate decreased GSH levels and increased MDA levels in the brain. The compound did not affect acetylcholinesterase (AChE) levels.

## 3. Beneficial Properties of Sodium Benzoate

As described above, many studies show that sodium benzoate, especially when taken in high doses, can pose a health risk through various action mechanisms. However, other studies indicate that this substance, through the same or similar mechanisms of action, has beneficial properties and may, in the future, be used in the development of therapeutic strategies for certain diseases.

### 3.1. Effects of Sodium Benzoate on Oxidative Stress and Inflammation

Since oxidative stress and inflammatory process accompanies many disease entities, antioxidant and anti-inflammatory activity is a desired activity for new drugs [77]. It is essential that such compounds act on specific antioxidant mechanisms present in a given disease entity. Thus, to achieve optimal therapeutic effects, it is necessary to find a compound that acts on several oxidative stress-related mechanisms present in a disease. The mechanisms of action of sodium benzoate are described below, as well as its implications in the treatment of selected disease entities.

Inflammatory responses of microglia and astroglia have been observed in various disease entities related to the nervous system. In lipopolysaccharide (LPS)-stimulated microglia cells, benzoate (>100–500 μm) suppresses NO production and decreases inducible nitric oxide synthase (iNOS) expression by inhibiting NFκB activation [78]. Furthermore, it inhibits the production and decreases the expression of TNF- α and IL-1 β. LPS increases the expression of MHC class II and B7-1 and B7-2 stimulatory molecules, and sodium benzoate counteracts these effects by suppressing their expression. In addition, the compound decreases CD11b expression (overexpression is associated with increased microglia activation) in microglia cells. It also inhibits LPS-induced activation of p21^ras^. It also affects astroglia cells, namely by decreasing the expression of iNOS but also by inhibiting the increased expression of glial fibrillary acidic protein (GFAP). The researchers suggest that the reduction in mevalonate pathway intermediates is likely responsible for the observed anti-inflammatory effects. Moreover, the compound reduces in vivo cholesterol levels in mice by 28% after 7 days of therapy. In another study, similar results were obtained, i.e., reduction in the proinflammatory cytokines TNFα and IL-6, as well as in cholesterol levels (sodium benzoate doses: 250, 500 mg/kg b.w.) [62].

Anti-inflammatory and antioxidant effects were also reported in a rat model of ulcerative colitis [79]. It (doses: 400 nad 800 mg/kg b.w.) decreased myeloperoxidase levels and increased GSH levels. In addition, it reduced bleeding, local edema, and tissue necrosis induced by acetic acid infusion. In a rat model of intracerebral hemorrhage, sodium benzoate (100 and 200 mg/kg b.w.) was effective in alleviating neurological deficits (improved long-term memory and spatial learning), maintaining the integrity of the blood-brain barrier (BBB), and reducing brain edema [80]. It also alleviated oxidative stress, as well as mitochondrial damage. In addition, it increased DJ-1 levels in mitochondria and exerted anti-apoptotic effects by increasing BCl-2 and decreasing Bax and caspase-3 and -9. It exerts these effects by participating in the DJ-1/Akt/IKK/NFκB pathway.

Furthermore, sodium benzoate affects the immune response by suppressing Th1-type responses and shifts the immune balance towards Th2 [56]. In the same study, the suppression of Th1 responses was found to occur through the inhibition of neopterin production and tryptophan degradation. The researchers suggest that the effects induced by benzoate may be beneficial to health in the context of inflammation while noting that, on the other hand, benzoate may promote allergies and reduce the effectiveness of the immune system in fighting pathogens. Neopterin is considered a marker of the immune response but is also thought to be produced by mycoglobin under stressful conditions [81]. Inhibition of its production may be a beneficial desirable effect in the treatment of inflammatory diseases.

Moreover, sodium benzoate is an inhibitor of amino acid oxidase (D-AAO). An in vivo study reported that this property may be responsible for its cytoprotective (renoprotective) effects [82]. In PC12 cells exposed to aluminum maltolate, benzoate was tested for its effect on cell survival [83]. This compound increased their survival and catalase levels but did not affect the levels of ROS and GSH. However, benzoate at higher concentrations showed cytotoxic effects. The cytotoxic effect was also observed in cancer cells. Sodium benzoate has also been shown to induce apoptosis and affect NFκB in these cells [84]. It should be emphasized that this compound inhibited colon cancer cell proliferation much more strongly than fibroblast cell proliferation.

### 3.2. Sodium Benzoate in Major Depressive Disorder and Anxiety

#### 3.2.1. Pathogenesis of Major Depressive Disorder

Major depressive disorder (MDD) is considered a civilization disease of the 21st century. It is believed that 280 million people suffer from MDD worldwide, and this number is still increasing yearly [85]. Despite the treatment, MDD often leads to suicide or the development of drug-resistant depression. MDD is a heterogeneous disorder; therefore, no single cause for its occurrence and progression has been found so far [86]. An important aspect of MDD is its low remission rate (complete remission only in about 50%) [87]. It is a disease associated with mood disorders and more specifically with mood decline. People with MDD also experience anhedonia, anxiety, sleep and appetite disturbances, feelings of worthlessness, and many other symptoms that are associated with long-term mood depression.

Hypotheses regarding the progression and formation of MDD are many. Some researchers have indicated a link between the NMDA receptor and MDD [88]. This receptor is a glutamate receptor, and any disruption in its balance can cause neurological or neuropsychiatric disease. Its function decreases with age, and this can be observed in learning and memory disorders. Antagonists of this receptor, e.g., ketamine, cause anti-depressive effects. For this reason, the compounds that modulate the NMDAR receptor function are being searched for. D-serine belongs to the co-agonists of this receptor. However, it has been shown that its long-term supplementation in transgenic mice can improve mood disorders [89]. Therefore, it seems reasonable to increase its concentration and, consequently, to inhibit its degradation by D-AAO by using inhibitors of this enzyme. It is commonly believed that in the course of MDD, the dopaminergic system is dysregulated, resulting in decreased dopamine levels, which in turn are responsible for some of the anhedonic behaviors [90]. 

#### 3.2.2. Animal Models Studies

Sodium benzoate injection has been shown to increase dopamine levels in the frontal cortex in rats [91]. Researchers suggest that the effect of sodium benzoate on cortical dopamine is due to local inhibition of D-AAO, resulting in decreased degradation of D-serine. Increased levels of this amino acid would act on the NMDA receptor, consequently stimulating dopamine neurons and dopamine release. Sodium benzoate (400 mg/kg and 800 mg/kg) was also tested in a rat model of MDD induced by chronic stress [92]. An improvement was observed in a sucrose preference test and a forced swim test. Moreover, sodium benzoate increased the levels of brain-derived neurotrophic factor (BDNF) and protein kinase A. The researchers suggest that this compound may alleviate the symptoms of MDD.

#### 3.2.3. Human Studies

In one case report, a patient with MDD was receiving only sodium benzoate (500 mg/day) [93]. He demonstrated an increase in thalamus, amygdala, and brainstem volume. This compound acted by inhibiting D-AAO, thus increasing D-amino acids. The symptoms decreased after just 2 weeks of administering benzoate, with partial remission occurring after 6 weeks. The researchers suggest that in addition to inhibiting D-AAO, this compound may prevent excitotoxicity-induced death and increase neuronal plasticity. It should be noted that the patient alone did not observe any side effects. Another similar case report also reported improvement in depressive symptoms after treatment with sodium benzoate (500 mg/day) [94]. In another case report, a patient who suffered from panic attacks received sodium benzoate for 6 weeks [95]. She suffered additionally from Parkinson’s disease (PD). This patient was shown to improve after taking this compound in the absence of side effects. The researchers suggest that this improvement may be due to an increase in D-serine levels due to the fact that benzoate belongs to the D-AAO inhibitors.

#### 3.2.4. Effects of Sodium Benzoate

Sodium benzoate was shown to reduce homocysteine levels in an animal [96]. High homocysteine levels are thought to be associated with the etiopathogenesis of MDD and anxiety [97,98]. Therefore, reducing its levels should contribute to reducing the symptoms associated with these diseases. However, homocysteine levels have not been measured in any case report or animal model of MDD or anxiety. In addition, sodium benzoate has been shown to have anti-inflammatory effects through its effect on the NFκB factor [78]. Its use in MDD therapy may be related to its cytokine theory, according to which there is an increase in pro-inflammatory cytokines but also activation of NFκB [99,100,101]. In addition, as previously mentioned, this compound inhibits tryptophan degradation and decreases neopterin production [56] and transiently increases anthranilic acid levels [8]. Increased tryptophan levels have been shown to be associated with decreased depressive symptoms [102,103], and anthranilic acid has also been shown to play an important role in the treatment of MDD [104,105]. In addition, high neopterin levels have been associated with the number of MDD episodes and also correlate with neuropsychiatric abnormalities [106,107].

Sodium benzoate was tested on mouse adipocyte cultures after LPS stimulation [108]. Under the influence of LPS, there was a decrease in leptin levels and an increase in IL-6 levels. Benzoate caused an even greater decrease in leptin levels in stimulated adipocytes, but did not affect IL-6 levels. Interestingly, low leptin levels were noted in women after a recent suicide attempt, which also suggests a link between the hormone itself and increased anxiety and overactivity of the hypothalamic–Pituitary–Adrenal (HPA) axis [109,110]. This link is confirmed by in vivo studies in rats in which elevated corticosterone levels were reported after benzoate administration [26]. Benzoate at a dose of 50 mg/kg b.w. did not affect these levels after one week, but after three weeks, elevated levels were already observed. Moreover, in higher concentrations, this compound already affected these levels after a week. Elevated corticosterone levels are closely related to the HPA axis. It should be noted that the effects of this compound on the human body may be cumulative due to the fact that the duration of exposure and its dose played a significant role in terms of hormone levels.

Ciliary neurotrophic factor (CNTF) is a neuronal survival factor [111]. Sodium benzoate was shown to increase its expression [112]. However, the correlation between CNTF and depressive and anxiety behaviors is quite unclear. Mice lacking this factor demonstrated behaviors similar to anxiety and depression [113]. In contrast, patients with MDD had elevated levels of CNTF compared to controls [114]. Other researchers suggest that in women, the inhibition of CNTF by progesterone reduces depressive symptoms, while in men, this factor has an antidepressant effect [115]. Perhaps this ambiguous effect of benzoate on depression may be related to sex hormones, due to the fact that benzoate affects the levels of both male and female sex hormones [40,116]. However, more research should be done in this direction because it is a compound with a high safety profile, and perhaps for some patients who poorly tolerate standard anti-anxiety or antidepressant therapy, it may be that sodium benzoate therapy can reduce such symptoms.

### 3.3. Sodium Benzoate in Schizophrenia

#### 3.3.1. Pathogenesis of Schizophrenia

Schizophrenia is one of the most severe mental diseases [117,118]. The first descriptions systematizing schizophrenia come from 1893. The symptoms of this disease can be divided into positive symptoms, negative symptoms, and cognitive disorders. Negative symptoms may include loss of function, anhedonia, decreased emotional expression, impaired concentration, and diminished social engagement [119]. As for positive symptoms, conceptual disorganization, delusions, and hallucinations are included. Negative symptoms predominate in one third of patients with this disease and are associated with poorer patient prognosis and response to treatment. The treatment of positive and negative symptoms includes drugs that affect dopaminergic, serotonergic, noradrenergic, and glutamatergic transmission [120,121]. Cognitive impairment is independent of the presence of positive and negative disorders in schizophrenia [122]. For example, there may be a decline in working memory, learning speed, reasoning, or problem solving. There are various hypotheses for the onset of schizophrenia [123]. These include the dopamine hypothesis, serotoninergic hypothesis, GABAergic (gamma-Aminobutyric acid) hypothesis, and glutaminergic hypothesis. Furthermore, it is believed that the pathophysiology of schizophrenia may involve the immune response and oxidative stress.

An important fact for many psychiatric disorders is that sodium benzoate belongs to D-AAO inhibitors [124]. This enzyme degrades D-amino acids in the process of oxidative deamination, among them D-serine and D-glycine, i.e., co-agonists of the NMDA receptor (antagonists such as phencyclidine (PCP) or ketamine exacerbate schizophrenia). According to one hypothesis, a decrease in the function of this receptor contributes to the development of schizophrenia. Furthermore, the expression and activity of this enzyme is elevated in patients with this disease [125]. It has been shown that benzoate changes the conformation of this enzyme to resemble a holoenzyme. Binding of the compound to D-AAO protects it from trypsinolysis and high temperatures. The effect of benzoate on the changes induced in this enzyme may be important for designing drugs for schizophrenia that affect D-AAO [126]. 

#### 3.3.2. Effects of Sodium Benzoate

Sodium benzoate, by inhibiting D-AAO, also increases the level of synaptic D-serine, which is a co-agonist of the NMDA receptor [127]. This causes an increase in the receptor excitability, which in turn is associated with the normalization of the receptor function (Figure 1). The reduced levels of D-serine are often associated with schizophrenia and bind more strongly to the NMDA receptor than D-glycine does [128]. As previously mentioned, sodium benzoate exerts an effect to increase d-glycine levels, and this amino acid is thought to be involved in the development of schizophrenia [129]. It was observed to have increased serum levels in patients with this disease. However, many researchers consider that by increasing its levels, symptoms of schizophrenia can be reduced. The above factors would suggest that benzoate could find potential use as a drug for schizophrenia because it increases the levels of two major NMDA receptor co-agonists.

Notably, the injection of sodium benzoate into rats increased extracellular dopamine in the frontal cortex [91]. The researchers suggest that it might be necessary to further investigate the mechanism of action of D-AAO inhibitors. Increased levels of dopamine in patients with schizophrenia contribute to the positive symptoms of the disease, and excessively low levels contribute to the negative symptoms. The action of inhibitors of this enzyme may also have a markedly different effect on the dopaminergic system than the drugs used for this disease, which act on dopamine receptors.

#### 3.3.3. Animal Models Studies

One animal model of schizophrenia uses PCP [130]. After its administration, the animals show hyperlocomotion (positive symptom index), deficits in social behavior in a social interaction test, and increased immobility in a forced swim test (negative symptom index), as well as deficits in sensorimotor gating and cognitive dysfunction in learning and memory tests. In this model of schizophrenia, sodium benzoate had antipsychotic effects [131]. A single dose (100, 300 lub 1000 mg/kg) of the compound alleviated pre-impulse deficits and hyperlocomotion in the animals. However, it is significant that a single dose of the compound did not affect D-serine levels in the plasma, frontal cortex, hippocampus, and striatum in mice. One limitation of this study is that D-serine levels were not measured in the cerebellum, wherein D-AAO levels are high, and only the effects of a single dose of benzoate were analyzed. In another study based on the same schizophrenia model, mice pretreated with benzoate (400 mg/kg) and D-serine demonstrated inhibition of locomotor activity after PCP treatment [132]. Interestingly, the administration of sodium benzoate did not increase the plasma and brain D-serine levels compared to the administration of D-serine alone. This compound decreased plasma L-serine levels without affecting brain L-serine levels. Sodium benzoate showed additive behavioral effects with D-serine. However, the researchers suggest that the mechanism of the beneficial effects of benzoate is not related to the inhibition of D-AAO but to its effects on the metabolism of small amino acids, such as L-serine and L-glycine. This mechanism is likely due to the fact that elevated levels of L-serine in patients with schizophrenia are usually observed [133]. The decrease in this level could be considered a response to benzoate treatment. The animal model of ketamine-induced schizophrenia includes positive symptoms but also negative symptoms, as well as cognitive symptoms [134,135]. In this rat model of ketamine-induced schizophrenia, sodium benzoate was compared with olanzepine [40]. A dose of benzoate 0.01 mg/kg for 10 weeks induced a slight increase in body weight and organ weights in rodents. This study also reported the effects of benzoate on liver parameters and sex hormone levels. Importantly, sodium benzoate had a beneficial effect on cognitive function, learning, memory, and spatial function.

#### 3.3.4. Human Studies

A clinical trial tested sodium benzoate as an adjunctive treatment for patients with schizophrenia who received sarcosine [136]. Sarcosine (an inhibitor of the glycine transporter) did not cause improvement in these patients. However, the combination of this compound with benzoate improved cognitive abilities in these patients. Benzoate as an additive was administered at a dose of 1 g/day for 12 weeks. Both compounds were well tolerated. It should be noted that these patients had previously received various therapies unsuccessfully. In addition, sodium benzoate (1000 mg/day) was administered as adjunctive therapy in patients with chronic schizophrenia who were receiving standard therapy for this disease [137]. It was shown that it could be effective as an adjunctive therapy in these patients. On the Positive and Negative Syndrome Scale (PANSS), the patients taking benzoate showed an improvement of up to 21% in total score. Their clinical symptoms decreased, and there was an improvement in neurocognition. They also had improved their processing speed and pattern memory. Again, the compound was well tolerated.

In another clinical study, sodium benzoate was administered to patients together with clozapine [138]. The results were consistent with the previously described study, i.e., patients showed improvement in the negative symptom scale but also in the PANSS. In addition, the compound was well tolerated in this study. It was tested in two doses i.e., 1 g and 2 g daily. The higher dose showed more beneficial effects in schizophrenic patients.

Similar to MDD, schizophrenia patients also have elevated homocysteine levels [139]. As previously mentioned, sodium benzoate decreases its levels [96], which may be one of the mechanisms for the beneficial effects of benzoate in this disease. Moreover, the compound contributes to the inhibition of NFκB activation, as previously mentioned [78]. Up-regulation of the NFκB signaling pathway has been observed in schizophrenia patients [140,141]. In addition, oxidative stress and excessive microglial activation are increased in this disease [142]. Thus, the beneficial effects of benzoate in schizophrenia are due in part to its ability to reduce oxidative stress and inhibit microglia activation [78,80].

Furthermore, low levels of tryptophan are reported in patients with schizophrenia [143,144]. Researchers suggest that the deficiency of this amino acid causes cognitive changes and also affects the integrity of the white matter in these patients. Moreover, high levels of neopterin are also noted in these patients [145]. As previously mentioned, sodium benzoate inhibits tryptophan degradation, resulting in increased tryptophan levels, and also inhibits neopterin production [56]. Thus, this may represent another mechanism by which this compound exhibits therapeutic effects in patients with schizophrenia.

In one study, sodium benzoate (1000 mg/day) showed no effect. It was tested as an adjunctive treatment for early psychosis [146]. The study on 100 participants reported no effect on the development of psychosis. Researchers consider that the lack of evidence for the compound’s effectiveness may be dictated by the fact that in other studies, patients developed long-term schizophrenia, or it was drug-resistant.

### 3.4. Sodium Benzoate in Neurodegenerative Diseases

#### 3.4.1. Pathogenesis of Neurodegenerative Diseases

Neurodegenerative diseases can include Alzheimer’s Disease (AD), PD, and multiple sclerosis (MS) [147,148,149,150]. These diseases usually lead to the degeneration of the nervous system by different mechanisms depending on the disease. For example, in AD there is an accumulation of tau protein and β-amyloid (βA) in the brain and acetylcholine (Ach) deficiency. Deposition of pathological proteins, decreased Ach levels, and increased oxidative stress and inflammatory response promote cognitive decline. AD is the most common type of dementia. These processes eventually lead to neuronal death. The disease progresses very rapidly. It is diagnosed only symptomatically, and its peak incidence is at age 65. The second most common disease after AD is PD [151,152]. It is caused by the degeneration of the dopaminergic system. Loss of dopaminergic neurons occurs mainly in the nigra pars compacta. In addition, α-synuclein aggregation occurs in this disease. Symptoms of PD include bradykinesia, resting tremor, and rigidity. The cause of MS is dysfunction of the immune system [153,154]. It is autoimmune in nature due to the autoantibodies present in patients with this disease. In addition, features of MS include demyelination, gliosis, and neuronal loss. The initial cause of MS is thought to be inflammation of white and gray matter tissues caused by the focal infiltration of cytokines, as well as immune cells. However, despite the slightly different causes of the neurodegenerative disorders, these diseases have many common features. These include oxidative stress, inflammatory response, and apoptosis. Microglia are also frequently activated.

#### 3.4.2. Effects of Sodium Benzoate in Neurodegenerative Diseases

Sodium benzoate does not only attenuate LPS-mediated microglia iNOS expression, as previously mentioned, but it also attenuates it after induction by other neurotoxins, such as βA (AD-related), MPP + (PD-related), and IL-12 p40 2 (MS-related) (Figure 2) [78]. Factors such as BDNF and neurotrophin-3 (NT-3) have protective functions for neurons and play an important role in alleviating neuronal loss in neurodegeneration [155]. Sodium benzoate increased the expression of BDNF and NT-3 in an in vitro study (cell culture), as well as in vivo in mouse brain cells. Researchers assume that this compound acts on these factors through the PKA CREB pathway. It is also significant that the compound easily penetrates the BBB. 

Additionally, benzoate inhibited ROS production in microglia cells induced by MPP+ (toxin in PD), LPS, and Aβ1-42 (AD etiological factor) [96]. This mechanism probably occurs through p21^rac^ inhibition, which was also confirmed in vivo (5XFAD Tg mice). The compound also reduced oxidative stress and gliosis in the hippocampus in animals. Moreover, the compound inhibited neurodegeneration in a mouse model of AD and reduced tau protein phosphorylation and βA levels. In addition, it affected learning and memory in mice. Moreover, depletion of the intermediates of the mevalonate pathway was shown to be responsible for the antioxidant effects of benzoate, as speculated in a previously mentioned study [78]. A reduction in homocysteine levels after sodium benzoate administration was reported in animals, as well [96]. The lowering of homocysteine levels by benzoate may be a beneficial feature for the treatment of other neurodegenerative diseases, as well, due to the fact that hyperhomocysteinemia is an independent risk factor for their occurrence [156,157].

In another study, sodium benzoate stimulated the expression of molecules associated with neuronal plasticity in mouse hippocampi, and it also improved the synaptic maturation of neurons in an in vitro study [158]. In cell cultures, the compound affected the calcium current of neurons, leading to enhanced cell membrane depolarization and, consequently, enhanced synaptic activity of neurons. What is more, the effect of benzoate on the PKA-CREB pathway was confirmed. Furthermore, it improved spatial learning and memory in poorly learning mice without affecting locomotor activity

It should be noted that CNTF administration is one of the therapeutic options for neurodegenerative diseases [111,159,160]. It is very likely that the beneficial effects of sodium benzoate in neurodegenerative diseases may be due to its effect on the up-regulation of CNTF expression and levels. In addition, it has an effect on inhibiting NFκB activation, as mentioned previously [78]. Moreover, this factor is a therapeutic target for neurodegenerative diseases [161]. Furthermore, benzoate inhibits the expression of p21 proteins [78,96]. They play a role in disorders such as Huntington’s Disease (HD) [162] and amyotrophic lateral sclerosis (ALS) [163]. As given above, sodium benzoate inhibits neopterin production [56]. Elevated levels of neopterin have been reported in AD patients and correlated with cognitive decline [164]. Increased neopterin levels were observed in PD [165] but also in HD [166]. It is probable that sodium benzoate may owe its beneficial effects on neurodegenerative diseases in part to its anti-inflammatory effect, as well as its action in inhibiting microglia activation through neopterin. Due to the above-described properties of sodium benzoate, more studies should be conducted on other neurodegenerative diseases.

#### 3.4.3. Effects of Sodium Benzoate on Alzheimer’s Disease

Sodium benzoate (250–1500 mg/day) was tested on patients with mild cognitive impairment, as well [167]. By assessing regional homogeneity (ReHo), how it affects brain activity, and more specifically local functional connectivity (FC), was examined. In the sodium benzoate-treated group, a positive correlation was noted between the change in non-verbal (spatial) working memory and ReHo in the right precentral gyrus and right middle occipital gyrus. In addition, in this group, a positive correlation was observed between memory and verbal learning and ReHo in the left precuneus.

In another clinical trial, benzoate (250–750 mg/day) was administered for 24 weeks to patients with mild AD and mild amnestic cognitive impairment [168]. This compound acted beneficially in improving cognitive function, as well as the patients’ general condition. The researchers attributed these changes to NMDA receptor stimulation. Benzoate was generally well tolerated. There were no changes in routine blood morphology, as well as biochemical tests from baseline assessment. Of relevance, sodium benzoate also inhibits tryptophan degradation [56]; tryptophan is reduced in AD patients, and low levels of tryptophan are responsible for cognitive dysfunction [169].

Moreover, benzoate has been shown to have beneficial effects in patients with behavioral and psychological symptoms of dementia [116,170,171]. Benzoate was administered at a dose of 250–1500 mg/day for 6 weeks. However, in this study, there was no elevation of D and L-amino acids after this therapy. In addition, catalase levels in the benzoate group showed an increasing trend but not significantly. On the other hand, cognitive function was shown to improve after benzoate treatment, which was correlated with decrease in D-AOO, female gender, younger age, higher BMI, and the patient’s baseline clinical status and their use of antipsychotics. There was also an increased ratio of estradiol to FSH in women, which may be related to modulation of the NMDA receptor by sex hormones after benzoate treatment.

#### 3.4.4. Effects of Sodium Benzoate in Parkinson’s Disease

Furthermore, some studies have undertaken evaluation of the therapeutic properties of benzoate in PD. Mutations in the DJ-1 protein are presumed to be associated with the pathogenesis and occurrence of PD [172]. It was mentioned previously that sodium benzoate in a mouse model of intracerebral hemorrhage had a beneficial effect on DJ-1 levels [80]. In another study, it increased the levels of this protein by modulating the mevalonate pathway in primary mouse and human astocytes, as well as in human neuronal cells [173]. Additionally, the compound increased the expression of other PD-related genes, such as Parkin, PINK1, LRRK2 and HtrA2, while suppressing the expression of α-synuclein. The researchers suggest that this compound could alleviate nigrostriatal damage in PD. In an animal model of PD (1-methyl-4-phenyl-1,2,3,6-tetrahydropyridine (MPTP) mouse model), sodium benzoate was also shown to affect levels of glial cell-derived neurotrophic factor (GDNF) [174]. It stimulated the increase in GDNF expression in mouse and human astocytes via CREB, but in vivo also had a beneficial effect on the levels of this factor in the substantia nigra pars compacta (SNpc). Thus, the compound improved locomotor activity in mice and also had a neuroprotective effect restoring the innervation of the striatum. It should be noted that the mouse model of MPTP PD has limitations [175]. One is that it is the effects of acute/post-acute administration of MPTP are spontaneously reversible over time. However, it is a good model for screening and also for studying the mechanism of action of the substance. In another study, sodium benzoate was shown to stimulate dopamine production by dopaminergic neurons by affecting the expression and protein levels of tyrosine hydroxylase [176]. As previously mentioned, locomotor improvement occurred in the animals. Disturbed trypofan metabolism is also observed in PD [165]. In these patients, there is an increased degradation of it. As mentioned above, sodium benzoate has a beneficial effect by inhibiting its degradation [56].

#### 3.4.5. Effects of Sodium Benzoate on Multiple Sclerosis

Shifting the balance of Th1 to Th2 is one of the therapeutic strategies in MS [177], and it was previously mentioned that sodium benzoate acted on both populations of these lymphophytes [56]. This was also confirmed by in vitro studies on peripheral blood mononuclear cells (PBMCs) in MS patients [178]. Mice with experimental allergic encephalomyelitis (EAE), a model in MS, were treated with sodium benzoate (2.5–10 mg/mL, and also, the mice drank water containing sodium benzoate-5 mg/mL) [179]. This compound inhibited the symptoms of the disease in both the acute and chronic phase. In addition, it reduced inflammation, thereby decreasing inflammatory infiltration, and it reduced demyelination and slowed disease progression. Furthermore, benzoate stimulated the production of Th2 cytokines, such as IL-4 and IL-10, while inhibiting the production of the Th1 cytokine viz: INF-γ and also increased Treg abundance. Cinnamon was tested in a mouse model of EAE, where it was shown to inhibit the disease process of EAE and reduce clinical symptoms [180]. It also had immunomodulatory effects (attenuated inflammatory infiltration and inflammation, inhibited the expression of pro-inflammatory molecules iNOS and IL1β, and affected Treg and Th17 populations), beneficial effects on BBB and blood–spinal cord barrier integrity, and inhibited demyelination. It also shifted the Th1 to Th2 balance. Perhaps cinnamon powder exerted this beneficial effect through its metabolite sodium benzoate. The important point is that the EAE model is considered a good model of MS because the model animals exhibit many mechanisms common to the course of MS [181]. This model mimics relapsing–remitting MS. One of the few differences is that the disease in these animals is induced artificially. CNTF induces neuronal survival and differentiation. It has neuroprotective, promyelinating, anti-apoptotic properties [111,182]. It has beneficial effects in neurodegenerative diseases. In MS, the level of CNTF decreases. The administration of sodium benzoate increased CNTF expression in astocytes and oligodendrocytes and also in vivo in the brain of EAE animals [112]. Thus, it was shown that benzoate may have beneficial effects in the treatment of MS due to its myelinogenic effects. Moreover, it was shown that benzoate requires protein kinase A to activate CREB and increase CNTF expression. Dysfunction and levels of Treg play a role in the pathogenesis and progression of MS [183]. Affecting this population may be one of the therapeutic strategies for MS. Importantly, benzoate affects this lymphocyte population by increasing TNF-β levels [184]. 

### 3.5. Sodium Benzoate in Pain Relief

Pain is a sign of a disorder [185]. According to some researchers, it should be considered a disease entity. There are two main types of pain: acute pain and chronic pain. The analgesic properties of sodium benzoate may be due to the fact that it belongs to D-AAO inhibitors [186]. It was shown that these inhibitors may be crucial in the treatment of chronic pain such as pain associated with bone cancer, but also neuropathic pain.

Sodium benzoate (400 mg/kg b.w.) is proved to relieve the pain in especially those associated with chronic pain [187]. It did not exhibit an effect in the acute phase of pain in an in vivo study in mice. Intraperitoneal as well as systemic administration of benzoate (400 mg/kg b.w.) to rats had analgesic effects in these animals [188]. Sodium benzoate reduced mechanical allodynia in rats with neuropathy and formalin-induced hyperalgesia. As before, no effect of the compound on acute pain was demonstrated. This effect was also confirmed in another study where benzoate prevented formalin-induced tonic pain in an in vivo model, yet had no effect on acute pain [189]. It also had a beneficial effect alleviating hypersensitivity to pain caused by sleep deprivation [190]. This effect was dose-dependent, and the maximum effect was observed 60 min after administration to rats.On the other hand, benzoic acid derivatives showed analgesic effects. These derivatives are believed to act as antagonists of prostaglandin E2 receptor subtype 4 [191]. Moreover, they acted specifically on adrenergic but also dopaminergic and GABAergic transmission [192]. Depressive disorders are very often associated with pain, as confirmed by many studies [193,194,195]. Chronic pain is usually related to an associated disease such as cancer. Severe pain makes it necessary for patients to take opiates, which cause numerous side effects. Since sodium benzoate has a high safety profile, it may be an alternative therapy for patients with chronic pain. However, it still requires research in this direction.

### 3.6. Sodium Benzoate in Autism Spectrum Disorder

#### 3.6.1. Pathogenesis of Autism Spectrum Disorder

ASD is a neurodevelopmental disorder likely due to early brain changes and neuronal reorganization [196,197]. Patients suffering from ASD vary from person to person. However, this term describes patients with a specific combination of disorders. These include impairments in social communication, repetitive sensory-motor behaviors, among others. There is a theory that patients with ASD have a disturbance in the immune system [198,199]. Moreover, such patients have NMDA receptor antibodies and other autoantibodies. Additionally, excessive activation of microglia and increased expression of CD95 on monocytes are observed in them. Increased pro-salivary cytokines and increased markers of T-lymphocyte activation are reported. In addition, the Treg/Th17 balance in patients with ASD is shifted toward Th17 [200]. Furthermore, a disturbed Th1/Th2 ratio is also observed [201]. As for the levels of D-serine or D-glycine, they are varied in these patients—both low and elevated levels have been reported [202]. 

#### 3.6.2. Effects of Sodium Benzoate

As mentioned above, sodium benzoate attenuates microglia activation (Figure 3) [78,96,155] and has a beneficial effect in shifting the balance towards Th2 [56,178,179]. Furthermore, it beneficially effects the Treg/Th17 ratio [180,184]. Therefore, these properties may support the potential use of sodium benzoate in the treatment of ASD patients. Another study administered sodium benzoate (for children with body weight ≥ 15 kg, the dose was 500 mg/day, for children with body weight < 15 kg, the dose was 250 mg/day) to non-communicative children with ASD for 12 weeks [203]. Half of the children demonstrated improvement in communication, and an activation effect was observed in three of the six children. However, the activation effect was not strong enough to require medical intervention or discontinuation of the study. In one case report, a girl with symptoms similar to ASD, but also with a urea cycle disorder, was administered thioridazine and also sodium benzoate (1.5–1.75 mmol/kg/day) [204]. Her condition improved after one year, and autistic symptoms were not noticeable. Perhaps the resolution of ASD symptoms was not only related to the use of thioridazine but in part to benzoate.

The beneficial effect of benzoate on patients with ASD may also be due to its effect on p21 expression, as previously reported [78,96]. Activated p21 kinase has been shown to be associated with severe ASD [205]. In addition, benzoate affects homocysteine levels [96]. Similar to the disorders described above, high levels are also observed in ASD [206,207]. It was mentioned earlier that sodium benzoate affects the inhibition of neopterin production and tryptophan degradation [56]. It should be noted that elevated levels are observed in individuals with ASD [208,209]. Abnormal tryptophan metabolism is also observed in these patients [210,211]. Given the effects of sodium benzoate in alleviating ASD as described above and other potential mechanisms of its action, more research in this direction would be warranted.

## 4. Summary

In summary, it should be emphasized that sodium benzoate is a compound with a broad safety profile and dose-dependent effects that are almost always adverse in the case of high doses. Many studies also indicate that sodium benzoate, through multiple mechanisms, has promising therapeutic potential, particularly for neurodegenerative diseases, ASD, schizophrenia, MDD, and pain relief. However, due to the fact that it is commonly present in food, the total dose taken by the patients cannot be completely controlled, and therefore, finding the right dose with the least possible side effects may present many difficulties. Further research is needed in this area.

## Figures and Tables

**Figure 1 nutrients-14-01497-f001:**
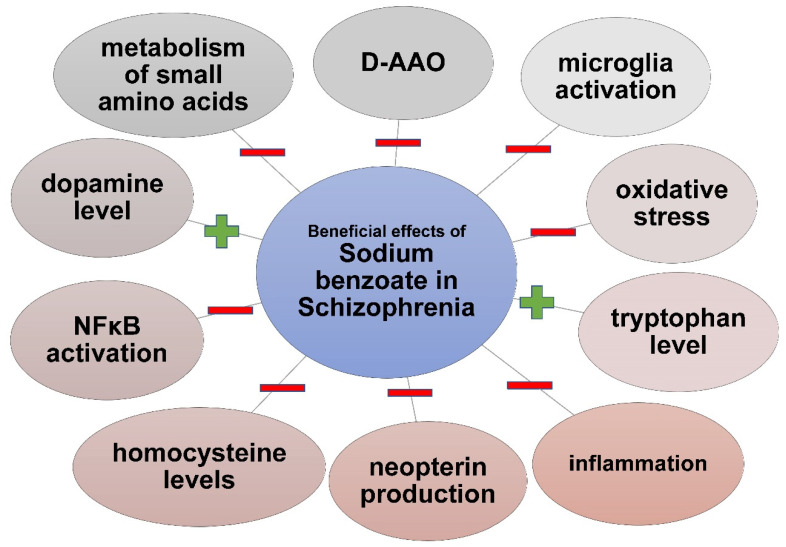
Summary of the beneficial effects of sodium benzoate in schizophrenia. ”−‘’ inhibition/reduction; “+’’increase.

**Figure 2 nutrients-14-01497-f002:**
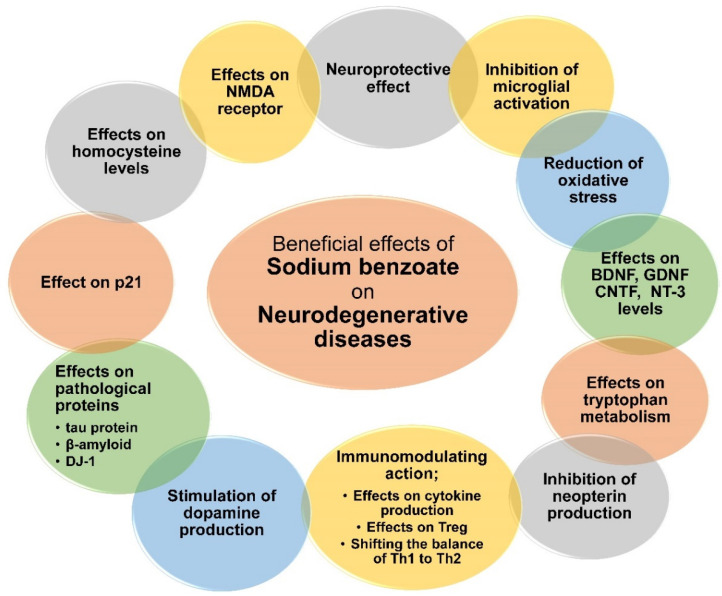
Summary the of beneficial effects of sodium benzoate on neurodegenerative disease.

**Figure 3 nutrients-14-01497-f003:**
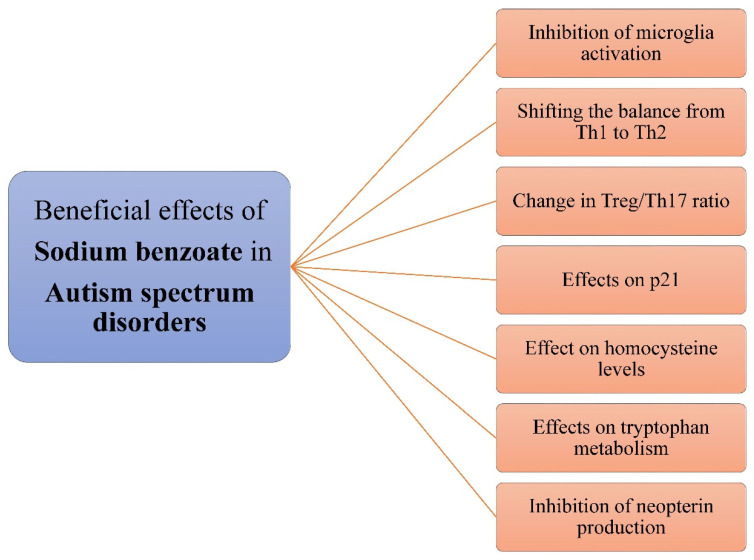
Summary of the beneficial effects of sodium benzoate on ASD.

## Data Availability

Not applicable.

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
