# Peer review of "Sodium Benzoate—Harmfulness and Potential Use in Therapies for Disorders Related to the Nervous System: A Review"

_nutrients, 2022, doi:10.3390/nu14071497_

Round 1

Reviewer 1 Report

Major comments:

First of all the authors need to define what kind of review it is: if this is the systematic review then the authors need to revise the manuscript according to PRISMA guidelines.

MDPI author instructions:

PRISMA covers systematic reviews and meta-analyses. Authors are recommended to complete the checklist and flow diagram and include it with their submission.

If this is a literature review the authors need to mention it in the title and to change some parts in the manuscript. The authors used many causative sentences in the manuscript which may be used in systematic review but one should be very careful using them in a literature review.

I would suggest the authors to change the title. For example: remove the term safety. You cannot conclude the safety of an ingredient or medicament in a review article. Even in many original articles, authors are very careful with such terms. As a result, authors need to revise the title (and mention if it is literature review or systematic review) as well as the whole text.

Authors mentioned in the manuscript that “literature from scientific databases was reviewed: Pubmed, Google Scholar and ScienceDirect”. If this is a systematic review, authors need to analyze and report the quality of the papers used in this manuscript.

Authors mentioned about Sodium benzoate effects in health and disease but they did not mention clearly in each section how this effect can lead to disease or health.

I would suggest the authors to reorganize the text, for example in schizophrenia section: Introduction, 1. Pathogenesis of schizophrenia, 2. Effects of sodium benzoate 3. Animal models studies 4. Human studies. This makes the text more understandable for the reader.

There is not enough information about animal models. For example in line 466. “In a mouse model of schizophrenia (after PCP treatment), sodium benzoate had antipsychotic effects”.  It is important to mention this is animal model of positive symptoms, because the drug could alleviated hyperlocomotion in the animals and if the other models (negative or cognitive) after sodium benzoate administration can show similar effects.

Some sentences in the manuscript lack the reference i.e. Line 731-732: In fact, the specific effect on the immune system of sodium benzoate undoubtedly makes it a desirable property when it comes to alleviating ASD (this is repeated again in line 732-734). Is it a conclusion? Indeed, it is very hard to conclude because sodium benzoate have some effects on immune system, as a result it can be efficient substance to alleviate ASD!!

Minor comment:

The figures quality is not good and it is hard to read the text inside the circles!

Author Response

Response to Reviewer 1 Comments

Point 1. First of all the authors need to define what kind of review it is: if this is the systematic review then the authors need to revise the manuscript according to PRISMA guidelines.

MDPI author instructions:

PRISMA covers systematic reviews and meta-analyses. Authors are recommended to complete the checklist and flow diagram and include it with their submission.

If this is a literature review the authors need to mention it in the title and to change some parts in the manuscript. The authors used many causative sentences in the manuscript which may be used in systematic review but one should be very careful using them in a literature review.

Response 1: We would like to thank the Reviewer for appreciating our work and for the suggestion, which helped us improve the quality of our manuscript. Our work is a literature review. In accordance with the comments and suggestion of the Reviewer, we mentioned it in the title and changed some fragments in the manuscript (mainly in the Abstract, in the Introduction and in the Research Methodology), which could indicate that it is a systematic review. All changes made are marked in red.

Point 2. I would suggest the authors to change the title. For example: remove the term safety. You cannot conclude the safety of an ingredient or medicament in a review article. Even in many original articles, authors are very careful with such terms. As a result, authors need to revise the title (and mention if it is literature review or systematic review) as well as the whole text.

Response 2: As suggested by the Reviewer, we have changed the title of the work - we have removed the word "safety" and added information that there is a literature review.

Point 3. Authors mentioned in the manuscript that “literature from scientific databases was reviewed: Pubmed, Google Scholar and ScienceDirect”. If this is a systematic review, authors need to analyze and report the quality of the papers used in this manuscript. 

Response 3: Our work is a literature review. The above sentence has been removed from the manuscript.

Point 4. Authors mentioned about Sodium benzoate effects in health and disease but they did not mention clearly in each section how this effect can lead to disease or health.

Response 4: In the revised version of the manuscript, we have added information about the mechanisms of action of sodium benzoate, through which it can have a harmful effect and through which it can positively affect some diseases (e.g. sections 2.1 and 3.1). We have also added 10 new references in the improved version of the manuscript.

Point 5. I would suggest the authors to reorganize the text, for example in schizophrenia section: Introduction, 1. Pathogenesis of schizophrenia, 2. Effects of sodium benzoate 3. Animal models studies 4. Human studies. This makes the text more understandable for the reader.

Response 5: As suggested by the Reviewer, the text was reorganized as indicated.

Point 6. There is not enough information about animal models. For example in line 466. “In a mouse model of schizophrenia (after PCP treatment), sodium benzoate had antipsychotic effects”.  It is important to mention this is animal model of positive symptoms, because the drug could alleviated hyperlocomotion in the animals and if the other models (negative or cognitive) after sodium benzoate administration can show similar effects.

Response 6: As suggested by the Reviewer, the manuscript has been supplemented with the missing information (section 3.3.3).

Point 7. Some sentences in the manuscript lack the reference i.e. Line 731-732: In fact, the specific effect on the immune system of sodium benzoate undoubtedly makes it a desirable property when it comes to alleviating ASD (this is repeated again in line 732-734). Is it a conclusion? Indeed, it is very hard to conclude because sodium benzoate have some effects on immune system, as a result it can be efficient substance to alleviate ASD!!

Response 7: We agree with this remark of the Reviewer. The indicated fragment has been removed from the manuscript. We added the following sentence: “Given the effective effects of sodium benzoate in alleviating ASD as described above and other potential mechanisms of its action, more research in this direction would be warranted”.

Point 8. The figures quality is not good and it is hard to read the text inside the circles!

Response 8: The quality of the figures has been improved.

Reviewer 2 Report

Comments:

This manuscript aims to assess the safety of sodium benzoate, and its potential use in the selected diseases, related to the nervous system by a literature review which was published between 1979 and February 2022 in Pubmed, Google Scholar, and ScienceDirect databases. This study analyses the properties of sodium benzoate, both in its harmful and beneficial effects, and its potential use in the treatment of nervous system diseases. They summarize the beneficial effects of sodium benzoate on schizophrenia, neurodegenerative disease and ASD in Fig 1-3. However, some harmful and beneficial results seem not to be consistent.

In addition, most referred references in this manuscript by the authors did not present the indented dose of sodium benzoate both in safety and treatment of nervous system diseases. As the conclusions of authors, “due to sodium benzoate is commonly present in food, the total dose taken by the patient could not be completely controlled, and therefore, finding the right dose with the least possible side effects may present many difficulties. “ I wonder the application of these results may be difficult and even impossible in the present form. Thus, the authors should try to integrate more organized, to help the readers to get the new and useful point.

Author Response

Response to Reviewer 2 Comments

Point 1: This manuscript aims to assess the safety of sodium benzoate, and its potential use in the selected diseases, related to the nervous system by a literature review which was published between 1979 and February 2022 in Pubmed, Google Scholar, and ScienceDirect databases. This study analyses the properties of sodium benzoate, both in its harmful and beneficial effects, and its potential use in the treatment of nervous system diseases. They summarize the beneficial effects of sodium benzoate on schizophrenia, neurodegenerative disease and ASD in Fig 1-3. However, some harmful and beneficial results seem not to be consistent.

Response 1: We would like to thank the Reviewer for appreciating our work and for the suggestion, which helped us improve the quality of our manuscript. Indeed, a literature review found that some studies were sometimes contradictory. We have added a sentence to the introduction that highlights the different conclusions of various scientific papers (line 67-70). In the revised version of the manuscript, we tried to clarify these discrepancies more clearly, for example by adding information on the doses of sodium benzoate at which it showed certain effects and by systematizing the experiments carried out, which we included in the review.

 Point 2: In addition, most referred references in this manuscript by the authors did not present the indented dose of sodium benzoate both in safety and treatment of nervous system diseases. As the conclusions of authors, “due to sodium benzoate is commonly present in food, the total dose taken by the patient could not be completely controlled, and therefore, finding the right dose with the least possible side effects may present many difficulties. “ I wonder the application of these results may be difficult and even impossible in the present form. Thus, the authors should try to integrate more organized, to help the readers to get the new and useful point.

Response 2: As suggested by the Reviewer, we have added information on the doses of sodium benzoate used in the research. We also tried to systematize the research described. We hope that the revised version of the manuscript is more understandable and provides valuable information on the topic under consideration.

Round 2

Reviewer 1 Report

The authors have addressed my concerns.

Reviewer 2 Report

The authors have revised the manuscript according to the suggestions of reviewers.